# Advances on Cellular Clonotypic Immunity in Amyotrophic Lateral Sclerosis

**DOI:** 10.3390/brainsci12101412

**Published:** 2022-10-20

**Authors:** Giuseppe Schirò, Vincenzo Di Stefano, Salvatore Iacono, Antonino Lupica, Filippo Brighina, Roberto Monastero, Carmela Rita Balistreri

**Affiliations:** 1Section of Neurology, Department of Biomedicine, Neurosciences and Advanced Diagnostics (BiND), University of Palermo, 90127 Palermo, Italy; 2Cellular and Molecular Laboratory, Department of Biomedicine, Neurosciences and Advanced Diagnostics (BiND), University of Palermo, 90134 Palermo, Italy

**Keywords:** Amyotrophic lateral sclerosis (ALS), neurodegeneration, neuroinflammation, neuromuscular disease, autoimmunity, the clonotypic immune system

## Abstract

Amyotrophic lateral sclerosis (ALS) is a fatal neuromuscular disease, characterized by the progressive degeneration of the upper and lower motor neurons in the cortex and spinal cord. Although the pathogenesis of ALS remains unclear, evidence concerning the role of the clonotypic immune system is growing. Adaptive immunity cells often appear changed in number, or in terms of their activation profiles, both peripherally and centrally; however, their role in ALS appears conflictive. Data from human and animal model studies, which are currently reported in the literature, show that each subset of lymphocytes and their mediators may mediate a protective or toxic mechanism in ALS, affecting both its progression and risk of death. In the present review, an attempt is made to shed light on the actual role of cellular clonotypic immunity in ALS by integrating recent clinical studies and experimental observations.

## 1. Introduction

Amyotrophic lateral sclerosis (ALS) is a neurological disease characterized by the irreversible and progressive loss of motor neurons located both at the cortical level—the so-called *first motor neurons*—in the gray matter of the spinal cord, and in the nuclei of the cranial nerves—the so-called *second motor neurons* [1]. Sporadic ALS accounts for about 90% of cases of the disease, whereas about 10% of ALS cases are hereditary due to genetic variants in numerous genes, including: the chromosome 9 open reading frame 72 (C9ORF72), Cu/Zn superoxide dismutase (SOD1), TAR DNA-binding protein 43 (TDP-43), and fused-in sarcoma/translocated liposarcoma (FUS/TLS) genes. These variations usually occur with autosomal dominant transmission (see Figure 1) [2]. The pathogenesis of both sporadic and familial forms is complex, partially clear, and certainly not restricted to a single factor (i.e., genetic factors may account for part of the pathogenesis, but they are also strictly related to several triggering and driving factors). The strong involvement of the immune system has been also documented. It is well recognized that the innate immunity plays a pivotal role in the central nervous system (CNS) and homeostasis, but also contributes to the onset of ALS. Accordingly, under physiological CNS conditions, innate immunity induces neuroinflammation to contain infections and eliminate pathogens, cell debris, and aggregated or misfolded proteins. With ALS, neuroinflammation is continuous and harmful for CNS cells; this constitutes the typical hallmark of the disease [3,4]. Moreover, in recent years, adaptive (or clonotypic) immunity [5] has emerged in studies concerning CNS health and disease; indeed, it is a fundamental component with a double function. First, it mediates immune-surveillance and defends against neurotropic viruses [6,7], as well as maintaining CNS homeostasis and integrity. It also promotes neurogenesis and improves cognitive function. With CNS degenerative diseases (i.e., ALS), clonotypic immunity commonly shows dysregulated and abnormal immune responses [4]. Accordingly, clonotypic immune cells often appear to be peripherally and centrally changed in several activation profiles. Data from human and animal model studies are currently reported in the literature. They provide the evidence on the capacity of each subset of lymphocytes, and their mediators, of caning mediate a protective or toxic mechanism in ALS, affecting both its progression and risk of death. In the present review, an attempt is made to shed light on the actual role of cellular clonotypic immunity in ALS by integrating recent clinical studies and experimental observations.

## 2. Recent Evidence on Clonotypic Immunity in ALS

The interplay between the clonotypic immune system and ALS emerges from the results of several clinical and experimental studies. For example, ALS patients are more often affected by autoimmune diseases [8], and the presence or absence of cognitive impairment in ALS patients has been associated with different peripheral immune profiles, with lower total lymphocytes, CD4+, B cell counts, and CD8+ lymphocytes in patients with cognitive decline than in those without objective cognitive impairment [9]. Current evidence on the biological effects, mediated by mutations in the abovementioned genes, additionally underlines a close relationship between the onset and progression of ALS and clonotypic responses of the immune system. Among them, mutations in the C9orf72 gene, which are particularly expressed in B cells and characterized by an expansion of the GGGGCC sequence within an intronic region [10], represent the most frequent cause of inherited ALS, and they indirectly reveal the close relationship between the disease and the active participation of the clonotypic immune system. Interestingly, C9orf72 knockout (KO) mouse models, with mild motor deficits, exhibit a dysregulated immune response, which is characterized by T-cell activation, overproduction of autoantibodies and cytokines, and signs of massive leukocyte infiltration, such as lymphadenopathy and splenomegaly, thus causing the development of a systemic lupus erythematosus-like disease [11]. C9orf72 KO mice also show the absence of mitochondrial degradation with Stimulator of Interferon Genes (STING) signaling, thus leading to the maintenance of interferon production and the activation of adaptive immunity [12]. Furthermore, mice harboring loss-of-function mutations in the ortholog of C9ORF72 develop splenomegaly, neutrophilia, thrombocytopenia, increased expression of inflammatory cytokines, and severe autoimmunity, ultimately leading to a high mortality rate [13]. Moreover, mice with full C9orf72 ablations show a typical induction of ALS pathological hallmarks, including motor neuron degeneration, gliosis, and increased ubiquitination, which is associated with a massive infiltration (detected by postmortem histopathological analyses) of histiocytes/macrophages and lymphocytes (particularly B220/CD45R-positive B-lymphocytes) in the CNS, as well as in the spleen, bone marrow, liver, kidneys, and lungs [14]. In addition, ALS patients with the C9orf72 mutation have higher levels of Interferon-α (INF-α) in their cerebrospinal fluid (CSF), when compared with ALS patients having other mutations [15] (see Figure 1). Interesting data on the relevant role of chronic inflammation in ALS also come from studies examining familial forms of the disease. These forms are evoked by FUS mutations, whose evidence suggests that the FUS protein may be a common component of clonotypic immune inclusions in non-SOD1 ALS [16]. Significant evidence concerning the role of clonotypic immunity and inflammation in ALS pathogeneses also emerges from the causative and susceptibility genes associated with the TDP-43 pathology, which are highly expressed in innate immune cells and increasingly implicated in key immune and inflammatory pathways, such as GRN and TBK1 [17].

In addition, it has been also reported that different subsets with different functions are involved, depending on the stage of the disease. T cells have been shown to enhance the survival of motoneuron cells (MNs) in Superoxide dismutase (SOD)1 mutant mice through protective neuroinflammation, which is likely to occur via Interleukin-4 (IL-4), and they infiltrate the spinal cord and brain in abundance during ALS progression [18]. Motor impairment has also been shown to be accompanied by a decline in the functions of regulatory T cells, which inhibits microglia activation in SOD1 mutant mice [19]. These data might suggest that neuroprotective functions of the immune system may prevail at an early stage of the disease, although other studies are needed to confirm this (see Figure 2). The progression of the disease is, however, accompanied by several changes in the immune system, such as: the acquisition of an inflammatory phenotype via microglia cells [20], thymic involution [21], increased levels of pro-inflammatory cytokines [22], and leucocyte infiltration into the central nervous system (CNS) [18]. In ALS, the infiltration of lymphocytes into the CNS has different consequences depending on the cell type. Accordingly, CD4+ CD25+ regulatory T cells (Tregs) and CD4+ T helper (Th)2 cells tend to mediate a neuroprotective effect, whereas the presence of CD4+ Th1, CD4+ Th17, cytotoxic CD8+, and Natural killer (NK) cells, and effector T lymphocytes (Teffs), is associated with a more rapid progression of ALS and an increased risk of death [22,23]. Few data are available on the role of B lymphocytes, plasma cells, and antibodies in the pathogenesis of ALS. Cases of paraneoplastic ALS with specific anti-neuronal antibodies deserve particular attention, because these are extremely rare forms of the disease; thus, definitively diagnosing these forms is difficult. However, the description of some of these forms in a recent review [24] suggests that there may be an interaction between cancer and neurodegeneration in ALS which is mediated by the immune system.

The evidence available in the literature for each of these cell types is described and discussed in this review.

## 3. T Helper 17 Cells (Th17)

Changes in the clonotypic cellular composition of the immune system have been found in ALS patients. A peripheral increase in the number of Th17 T cells has been found to be positively correlated with ALS symptom severity and disease progression [25]. In addition, patients with ALS have been shown to have increased levels of Interleukin-17 (IL-17), in both the serum and CSF [26]. Furthermore, higher levels of IL-17 have been quantified in ALS patients when compared with patients with primary progressive multiple sclerosis (PPMS) [27]. Among the Th17-related cytokines, only IL-17A has been shown to have a clear pathogenetic role in ALS models. Indeed, MNs of patients with ALS, derived from the differentiation of induced pluripotent stem cells, were found to express the receptor for IL-17A (IL-17AR), and they were vulnerable to its neurotoxic action in a dose-dependent manner; however, they were not damaged by exposure to IL-17F. Furthermore, targeting IL-17A has been shown to protect MNs from death [25]. Similarly, Th17 and IL-17A were shown to impair MNs survival in a FUS-related ALS mutant model [28]. IL-17 secreting cells, CD8+ T cells, and mast cells, have been observed to infiltrate the gray matter of the spinal cord in ALS patients. In these subjects, increased peripheral levels of IL-17A have been reported to mirror the decreased serum levels of IL-10, which has anti-inflammatory effects. This could help explain the increased susceptibility to the effects of IL-17A [29].

## 4. CD8+ T Cells

CD8 + cells have been shown to be activated both peripherally and intrathecally in ALS patients, compared with healthy controls, dementia patients, and PPMS [30]. A high percentage of CD8+ lymphocytes has been found to be negatively associated with ALS prognosis, and its increase has been correlated with the risk of death [31]. Although it was initially reported that the infiltration, of the spinal cord by CD8+ lymphocytes, occurs in the later stages of the disease, in a mouse model, a recent work has also revealed their presence in the early stages of the disease [32]. CD8+ lymphocytes infiltrating the spinal cord of a SOD1^G93A^ mouse model of ALS have been shown to interact with MNs through the major histocompatibility complex class I (MHC-I), and they induce killing by involving Fas and granzyme mechanisms [33]; however, the role of CD8+ lymphocytes in ALS, mediated by their interaction with MHC-I, has proven to be rather complex and still unclear. Although CD8+ lymphocytes have been shown to be toxic for spinal cord MNs in ALS models, some protective effects have been observed at a more peripheral level. More specifically, using a SOD1^G93A^ mouse model, the absence of the interaction between the sciatic nerve, MHC-I on the surface of the motor axon, and CD8+ lymphocytes appears to accelerate atrophy and the denervation of hindlimb muscles, thus indicating the onset of disease symptoms; however, it has been shown that the lack of interaction between CD8+ lymphocytes and microglia in the spinal cord protects cervical MNs from death and it delays disease onset [34].

## 5. Natural Killer (NK) Cells

The presence of NK cells, which contribute to innate and adaptive immunity, was observed in the spinal cord, and the motor cortex of postmortem tissues, of sporadic ALS (sALS) patients, whereas the presence of these cells was not found in the tissues that were used as controls. In contrast, NK cell levels were shown to be reduced in the peripheral blood of sALS patients compared with controls [35,36]. Similarly, postmortem data in humans and SOD1^G93A^ mice have reported that NK cells infiltrate the motor cortex and the spinal cord with a peak concentration occurring during the early phase of the disease, and a decreased number during motor decline. In addition, it has been observed that such mice express high levels of activation markers on their cell surfaces. In SOD1^G93A^ and TDP43^A315T^ mice, early treatment with anti-NK cell antibodies has been shown to increase survival and delay the onset of the disease [36]; however, NK cell depletion has been shown to prolong survival in female, but not male, SOD1 mice, thus suggesting that NK cells are related to ALS in a sex-specific manner [37]. In addition, it is also important to consider the subtype of NK cells rather than their total number. Indeed, although some authors noted no change in the total number of NK cells in the CSF of their study patients compared with controls, cell characterization in the slow-progression ALS group showed an increase in the number of regulatory, rather than cytotoxic, NK cells [25]. In general, the immune aspects of NK cells in ALS are poorly explored, and further investigations are needed given the complexity of the topic, as shown by a possible paraneoplastic case of MNs in the disease, with a rapid progression that is associated with NK cell leukemia [38].

## 6. Regulatory T (Treg) Cells

It has been reported that the CD4+ CD25+ Treg levels correlate with the rate of disease progression in ALS patients and mice models. Using the Appel ALS score (AALS), 54 patients with ALS were clinically evaluated, and they were divided into two groups according to disease progression. The 28 patients with a slowly progressing version of the disease (AALS points per month <1.5) had a percentage of Treg lymphocytes that did not vary from the controls. In contrast, the 26 patients with a rapidly progressing version of ALS (AALS points per month ≥1.5) showed a percentage of Treg cells that were reduced by about one-third compared with controls and patients with a less aggressive version of the disease [39]. In another study from Sheean and colleagues, the levels of Treg lymphocytes in 24 male and 9 female ALS patients were found to be inversely correlated with the rate of progression, although no clear difference was found between the Treg cell levels of ALS patients and those of the control group [40].

In Cu^2+^/Zn^2+^ SOD1 mutant (mSOD1) ALS mice, an increase in the number of Treg lymphocytes was found in the early stages of the slowly progressing version of the disease. This has been shown to improve the course of the disease as the expression of IL-4 is increased and the protective role of M2 microglia is enhanced. In addition, the transfer of CD4+ T lymphocytes from SOD1 mutated mice, with an increased number of Tregs, to SOD1^−/−^ mice, has been shown to reduce disease progression when compared with the transfer of wild-type CD4+ T lymphocytes [41].

Thus, the course of ALS appears to be characterized by an initial phase in which the neuroprotective role of the immune system dominates, and a second phase that is characterized by the development of neurotoxicity via microglia and proinflammatory Teffs [22]. Furthermore, it has been observed that Treg cells support the neuroprotective phase of the disease by inhibiting microglia through the production of IL-4, and Teffs through the production of IL-4, IL-10, and Transforming Growth Factor-β [19].

## 7. B-Cells and Immunoglobulins

Although some authors initially suggested that that B-cell, and even anti-retroviral immune responses, may be present in ALS [42], few data are available on the involvement of B cells in ALS, and in any case, their role appears to be very limited. B cells isolated from SOD1 mouse models of ALS before, during, and after the onset of disease showed a phenotype and responsiveness that was like those of the wild-type mice. In addition, the SOD1 mice that were lacking mature B cells, because they were blocked at the pro-B cell stage, were shown to develop the disease with clinical features that were identical to those of the control SOD1 mice [43]. It has been shown that possible signs of B cell involvement in some mechanisms of the disease arose mostly from indirect signals. Indeed, autoantibodies against neurofilaments, actin, and desmin have been found in the spinal cord of ALS patients, and these antibodies have been shown to positively correlate with disease severity [44]. In contrast, the presence of anti-SOD1 antibodies has shown a positive association with survival in sALS patients [45]; however, it should be noted that although T helper lymphocytes and cytotoxic T lymphocytes infiltrate the areas affected by degeneration, no infiltration by B lymphocytes has been found in the tissues of ALS patients [46], although the expression of the IgG subclass in ALS has been found to be altered [47], which suggests some dysfunction in the B lymphocytes.

Two animal models of autoimmunity, developed for ALS study, were used to explore the role of IgG reactivity in ALS. The first involves experimental autoimmune motor neuron disease that was induced via the inoculation of purified MNs and characterized by the loss of lower MNs. The second involves experimental autoimmune gray matter disease (EAGMD), which was induced via the inoculation of homogenate from the ventral horn of the spinal cord, thus leading to the death of the upper and lower MNs. Both models mimic ALS with respect to the depletion of MNs in terms of their neurophysiological findings [48]. The passage of serum immunoglobulins, which were isolated from both models and transferred to control mice, appears to passively reproduce some alterations in the animal-derived models, such as increased calcium levels within MNs and the increased release of acetylcholine from the axons of spinal MNs; the latter alteration suggests a possible role of an antibody-mediated response in ALS, with respect to interactions concerning the neuromuscular junction [48,49]. In ALS, the IgG accumulation in MNs has been described as inducing altered calcium homeostasis, and the inoculation of anti-MNs antibodies induces similar alterations in mice [50]. Similarly, Polgár and colleagues observed that serum transfers from ALS patients with the C9orf72 mutation into ventral spinal cord mice causes increased calcium levels in MNs, thus resulting in neurodegeneration [51]; similar data were provided by Obál and colleagues through the long-term intraperitoneal injection of sera from ALS patients [52].

## 8. Immunotherapy in ALS: Promising Data

Promising results have been reported in the literature concerning the potential role of the clonotypic immune system in negatively influencing the course of ALS. This has led to the development of substances which use immunomodulators, and which produce biological effects of particular interest. Accordingly, some authors have shown that Treg cell dysfunction is transient, and their expansion, under different environmental conditions, can lead to a recovery of their functions. Subsequent autologous transplantations of these cells have also been shown to slow the progression of sALS in a phase I clinical trial [53]; therefore, such approaches seem promising. In Table 1 below, the main immuno-modulatory agents and treatment options, that have been examined in ALS patients and animal models thus far, are shown. They show interesting results and may also encourage further studies on larger cohorts to validate their effects and find any possible adverse reactions. They may also elucidate the actual role of clonotypic cells in ALS. This could enable the development of more appropriate treatments for preclinical and clinical stages of ALS which could delay or halt its onset and progression. In addition, works based on omics technologies that are used to characterize and detect the clonotypic subsets in ALS, as well as its onset and progression stages, could enable the development of personalized therapies. Overall, if confirmed, these research hypotheses will play an important role in terms of ALS treatment and prognosis, with significant economic implications for national health care systems.

## 9. Conclusions

Recent data suggest that ALS, the classic neurodegenerative disease, usually having a rapid progression, is characterized by inflammation, which may also involve cells of clonotypic immunity, in addition to glial activation, and consequently, the induction of innate immunity [62]. At present, the early pathogenetic role of such cells cannot be hypothesized, even if it has been shown in an animal model of SOD1-ALS T-lymphocytes, that such cells infiltrate the CNS before the onset of symptoms [63]; however, the balance between CD4+ and CD8+, and between Treg and Teffs cells, has been reported to influence the prognosis of the disease (see Figure 3). In addition, there is no clear association between the dysfunctions of the immune system, particularly those regarding the clonotypic immunity, and the different clinical forms of ALS. The higher incidence of autoimmune diseases, such as myasthenia gravis, polymyositis, dermatomyositis, and type I diabetes mellitus [8] in ALS patients, suggests an increased reactivity of the immune system. We speculate that the progressive neurodegeneration causes the release of autoantigens and triggers the antibody-mediated response. In contrast, other authors have hypothesized that the formation of aggregates by SOD1 and TDP-43, which is triggered by CD4+ cells, may elicit an immune response in the CNS towards certain antigens [62]. Certainly, other studies are needed for clearing such relevant aspect, by facilitating the development of related treatments. 

## Figures and Tables

**Figure 1 brainsci-12-01412-f001:**
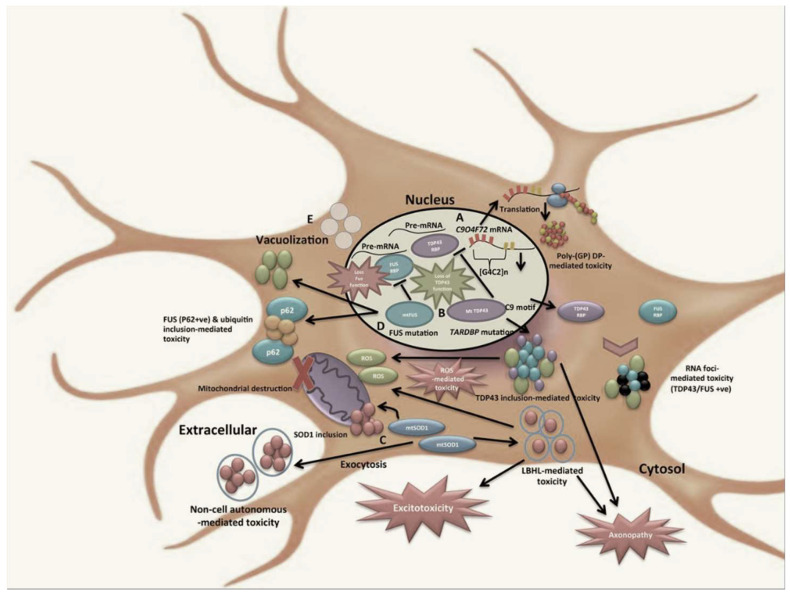
Pathophysiological effects induced by *C9OR72, TARDBP, SOD1, and FUS* gene mutations that are associated with the onset of ALS. A: The C9ORF72 mutation induces a gain-of-function (GOF) mechanism. More precisely, GGGGCC (G4C2), when translocated to its cytosol form, aggregates poly-(GP) dipeptide-repeat proteins (DPR), or misfolded proteins, to produce aggregates of ubiquitinated (U) RNA foci that are associated with TDP43 or FUS proteins, which are both mediating neuronal toxins. B: The transactive response of the DNA-binding protein (TARDBP) mutation triggers both the loss of function (LOF) and GOF mechanisms. C: The superoxide dismutase 1 (SOD1) mutation provokes a GOF mechanism. Mutant SOD1 dimers in the cytosol collect as SOD1 inclusions within mitochondria, and Lewy-body-like hyaline (LBLH) inclusions in the cytosol, where they can trigger mitochondrial reactive oxygen species (ROS) a generation later, thus causing mitochondrial destruction. D: The fused-in sarcoma (FUS) mutation mediates both LOF and GOF mechanisms. Mutant FUS proteins cause LOF by preventing normal FUS from binding to pre-mRNA. E: Cytosol vacuolization is caused by all the above-mentioned mutations. **By Biorender software**.

**Figure 2 brainsci-12-01412-f002:**
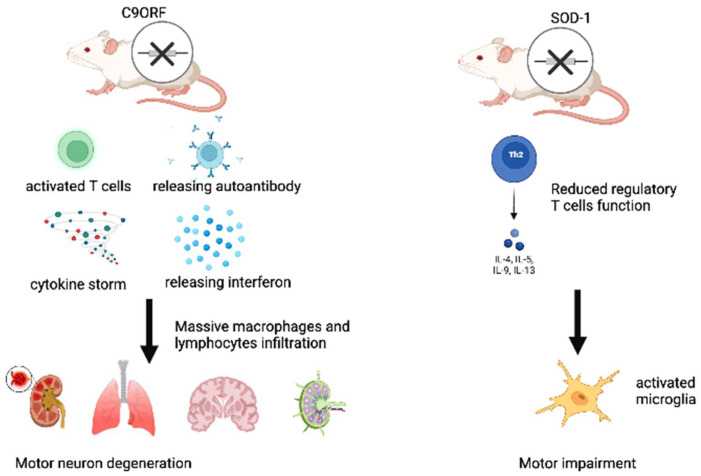
*C9ORF* and *SOD1* mouse models which are mainly used for providing evidence concerning the involvement of the clonotypic immune system in the pathogenesis of ALS. **By Biorender software**.

**Figure 3 brainsci-12-01412-f003:**
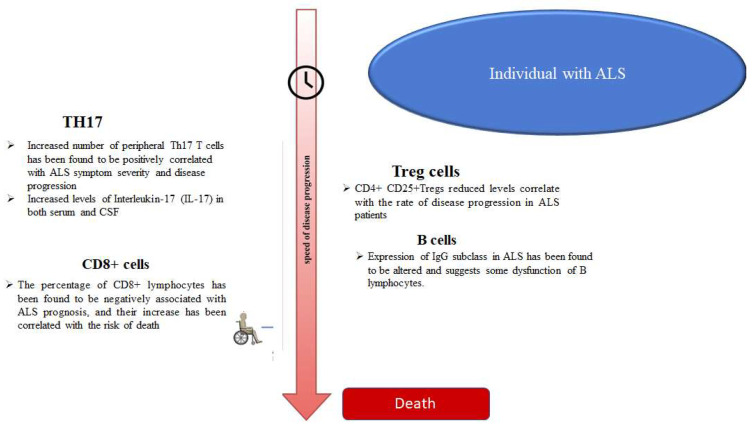
Correlation between the number and levels of clonotypic cells and related cytokines in terms of disease progression and outcomes. **By Biorender software**.

**Table 1 brainsci-12-01412-t001:** Main immune-modulating agents and treatment options that have been evaluated in ALS patients and in animal models.

Treatment	Dose and Administration	Number of Cases or Types of Animal Model	Laboratory and/or Clinical Outcomes	References
Aldesleukin	Intravenous Low dose of Interleukin-2 for five days at week 1, 5, and 9.	Twenty-four patients	Increase in Treg response; no significant change in terms of a reduction in circulating lymphocytes in ALS patients when compared with the placebo group.	[54]
Dimethyl fumarate	Oral administration of 480 mg daily for 36 weeks.	Seventy-two patients	No significant effects in terms of a reduction in circulating lymphocytes in ALS patients compared with the placebo group.	[55]
RNS60	Weekly intravenous infusion and daily nebulization.	Thirteen patients	No changes in biomarkers (i.e., FOXP3 mRNA and IL-17 levels).	[56]
	300 μL/mouse intraperitoneally every other day.	C57BL/6-SOD1G93A	Increase in CD4+/Foxp3+ T regulatory cells and neuroprotection.	[57]
	Three hundred and seventy-five mL intravenously administered for 24 weeks, once a week, and 4 mL/day administered via nebulization on the other days.	Seventy-two patients	Slower decline of forced vital capacity and bulbar dysfunctions in the patients treated with RNS60 compared with the placebo group.	[58]
Infusions of autologous Treg in ALS	Intravenous Tregs 106 cells/kg, initially four doses over 2 months, and in later stages, four doses over 4 months of the disease (n. 8 total infusions).	Three patients with sALS	Increase in Treg function and a slower disease progression, measured in accordance with the Appel ALS scale for each patient.	[53]
Fingolimod	Oral administration, dose of 0.5 mg/day for 4 weeks.	Eighteen patients with ALS	Reduction of circulating lymphocytes in ALS patients. No effects on ALSFRS-R.	[59]
Glatiramer acetate	Subcutaneous injection of 40 mg/day.	Three hundred and sixty-six patients with ALS enrolled in a phase II clinical trial	No effects on ALSFRS-R.	[60]
Tocilizumab	Treatment of PMBCs overnight with 2 μg/mL apo-G37R + 10 μg/mL tocilizumab.	Four patients with ALS	Reduction of the secretion of cytokynes and chemiokynes from the PMBCs of patients.	[61]

## Data Availability

Not applicable.

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
