# Peer review of "Advances on Cellular Clonotypic Immunity in Amyotrophic Lateral Sclerosis"

_brainsci, 2022, doi:10.3390/brainsci12101412_

Round 1

Reviewer 1 Report

Growing evidences have proved the important role of clonotypic immune system in the pathogenesis of ALS. The paper integrated the recent clinical studies and experimental observation, improving our understanding of clonotypic immunity in ALS. However, there are still some places that need to be revised in the paper as far as I am concerned.

1. When the abbreviations were first used, the full names should be given (e.g., th17). Please check if there are any and grammatical and spelling mistakes? (e.g., line 42, “as well as ASL”? line 56, “ASL”?)  MN is mononuclear cells or motor neuron? And you should keep the same font style (SOD1 G93A or SOD1G93A?). 

2. Line 84-85: “these data suggest that the neuroprotective functions of the immune system may prevail in an early stage of the disease”. I think the given information can’t support this conclusion adequately. You should give more detailed evidences before this conclusion.

3. Line 106-107: “Interleukin-17 (IL-17) levels were assessed increased in both serum and CSF of ALS patients, likely indicative of Th17 activation.” Who were compared with ALS patients in IL-17 levels? Healthy controls? Line 107-109: “higher levels of IL-17 have been quantified in ALS patients compared with patients with primary progressive multiple sclerosis.” Are these two sentences repeated?

4. The description about the immunotherapy in the last part can be separated as an independent section. The conclusion part can be placed last and be concise.

Author Response

Growing evidences have proved the important role of clonotypic immune system in the pathogenesis of ALS. The paper integrated the recent clinical studies and experimental observation, improving our understanding of clonotypic immunity in ALS. However, there are still some places that need to be revised in the paper as far as I am concerned.

1.Author comment. We thank the Reviewer 1 for the positive evaluation about of our manuscript, and for his/her important suggestions

  1. When the abbreviations were first used, the full names should be given (e.g., th17). Please check if there are any and grammatical and spelling mistakes? (e.g., line 42, “as well as ASL”? line 56, “ASL”?) MN is mononuclear cells or motor neuron? And you should keep the same font style (SOD1 G93A or SOD1 ?).

2.Author comment. As suggested by the Reviewer 1, we revised the abbreviations into the text of our manuscript

  1. Line 84-85: “these data suggest that the neuroprotective functions of the immune system may prevail in an early stage of the disease”. I think the given information can’t support this conclusion adequately. You should give more detailed evidence before this conclusion.

3.Author comment. As underlined by the Reviewer 1, we rearticulate the sentence

  1. Line 106-107: “Interleukin-17 (IL-17) levels were assessed increased in both serum and CSF of ALS patients, likely indicative of Th17 activation.” Who were compared with ALS patients in IL-17 levels? Healthy controls? Line 107-109: “higher levels of IL-17 have been quantified in ALS patients compared with patients with primary progressive multiple sclerosis.” Are these two sentences repeated?

3.Author comment. As underlined by the Reviewer 1, we revised the entire text and improved all the sections

  1. The description about the immunotherapy in the last part can be separated as an independent section. The conclusion part can be placed last and be concise.The reviewer addressed most of my concerns.

4.Author comment. As underlined by the Reviewer 1, we revised such sections, and created two diverse paragraphs.

Reviewer 2 Report

This is an important paper in an attempt to better understand  the ALS pathogenesis.

This is an important and even not known theme.

Author Response

This is an important paper in an attempt to better understand the ALS pathogenesis. This is an important and even not known theme.

1.Author comment. We thank the Reviewer 2  for the positive evaluation about of our manuscript, and for his/her important suggestions

Reviewer 3 Report

The present review aims to summarise the role of lymphocytes and their mediators in ALS. The review is clearly written and well organised.

Few suggestions are listed below to improve the paper.

- Lines 68-72: “Interestingly, C9orf72 knockout (KO) mouse models, with 68 mild motor deficits exhibit a dysregulated immune response, characterized by T-cell activation, overproduction of autoantibody and cytokines, and signs of massive leukocyte infiltration, such as lymphadenopathy and splenomegaly, developing a systemic lupus erythematosus-like disease (11).” The authors should cite also the other models published for C9orf and support their ideas (Loss of function from Burberry et al. 2016, full ablation from Sudria-Lopez, 2016, etc,). The authors could try to briefly explain where C9orf is known to be expressed (eg in B cells)

 -       Developing a bit more the link between different genetic form of ALS and inflammation could help to have a complete overview (eg. role of inflammation in other murine models: FUS etc).

-       The authors could try to propose a schema summarizing the key points between the lymphocytes and cytokine levels, disease progression and ALS patient outcome

Minor:

- Ref format does not seem correct: need [] instead of ()

- Line 56: type: should be “ALS” instead of “ASL”

- Line 94: space missing “(19; 20).Few”

- Table 1: row for ref 57: “Riduction of circultaing lymphocytes in ALS patients. No effects on ALSFRS‐R” need to change to “Reduction of circulating lymphocytes in ALS patients. No effects on ALSFRS‐R”

Author Response

 The present review aims to summarise the role of lymphocytes and their mediators in ALS. The review is clearly written and well organised.

1.Author comment. We thank the Reviewer 3  for the positive evaluation about of our manuscript, and for his/her important suggestions

Few suggestions are listed below to improve the paper.

- Lines 68-72: “Interestingly, C9orf72 knockout (KO) mouse models, with 68 mild motor deficits exhibit a dysregulated immune response, characterized by T-cell activation, overproduction of autoantibody and cytokines, and signs of massive leukocyte infiltration, such as lymphadenopathy and splenomegaly, developing a systemic lupus erythematosus-like disease (11).” The authors should cite also the other models published for C9orf and support their ideas (Loss of function  from Burberry et al. 2016, full ablation from Sudria-Lopez, 2016, etc,). The authors could try to briefly explain where C9orf is known to be expressed (eg in B cells) -

2.Author comment. As suggested by the Reviewer 3, we revised the sentences indicated and added other data related to the evidenced studies.

Developing a bit more the link between different genetic form of ALS and inflammation could help to have a complete overview (eg. role of inflammation in other murine models: FUS etc).

3.Author comment. As suggested by the Reviewer 3, we improved the sections indicated, as well as  all the sections of our manuscript, by better emphasizing the author message and making clearer the reading flow.

 - The authors could try to propose a schema summarizing the key points between the lymphocytes and cytokine levels, disease progression and ALS patient outcome

4.Author comment. As suggested by the Reviewer 3,we summarized the key points about the correlation between number of immune cells and levels of cytokines and ALS progression and outcomes in a clear figure

Minor: - Ref format does not seem correct: need [] instead of () - Line 56: type: should be “ALS” instead of “ASL” - Line 94: space missing “(19; 20).Few” - Table 1: row for ref 57: “Riduction of circultaing lymphocytes in ALS patients. No effects on ALSFRS‐R” need to change to “Reduction of circulating lymphocytes in ALS patients. No effects on ALSFRS‐R”

5.Author comment. As evidenced, we improved the indicated sentences and Table.

Round 2

Reviewer 1 Report

The authors have made revison according to the comments, and added some figures, which improve the paper quanlity well.  Please check the spelling and grammar carefully.